# OpenReview forum: "Phishing Vs. Legit: Comparative Analysis of Client-Side Resources of Phishing and Target Brand Websites"
_ACM.org/TheWebConf/2024/Conference — TheWebConf24 Oral_

### Official Review · Reviewer_UqMr · 2023-11-21

**Novelty:** 5
**Technical Quality:** 5

**Review:**

- This paper is a study on how client-side resources (with a focus on JavaScript libraries) are used in phishing websites by comparing them with resources used in legitimate target websites. The work tries to answer some research questions such as the types of prevalent libraries in phishing websites, and HTML structure similarities between phishing and legitimate websites. This is done by measuring the client-side resources of both phishing websites and the corresponding legitimate target websites in a systematic manner over data collected for 25 months. The work reveals that phishing websites tend to employ more diverse JavaScript libraries than their legitimate counterparts, and some libraries such as Socket.IO are typically only used in phishing websites.

Strong Points:
- This work conducts a longitudinal study / comparative analysis of phishing websites and legitimate target websites in order to gain insights on client-side resources that have not been detected before (I do have some questions about this, see my comments below). But all in all, the findings in this work are interesting and even allows for phishing mitigation suggestions.
- Most the data collection / resource collection processes are time-consuming (ex. identifying legitimate target brands through comparing screenshot appearances, collecting client-side resources of legitimate target brand websites using the Wayback Machine), so studies like these provide valuable insights for future research.

Weak Points:
- Parts of the paper are somewhat hard to follow, and may mislead the readers. For example, in the abstract: "...we find that a considerable portion of them still maintain a basic and simplistic structure, while phishing websites have significantly evolved to bypass anti-phishing measures, such as 2FA", this sentence is somewhat misleading in short glance that the legitimate services are basic and simplistic as compared to phishing websites, instead of the phishing websites themselves. I suggest that the authors rephrase such sentences to make things more clear for the reader.

**Questions:**

- The related work section does mention some previous works that have measured client-side resources on benign websites and malicious websites. Are all contributions listed in this paper novel? In other words, were previous works not able to find the insights revealed in this work?

- The authors have pointed out that each cluster was manually inspected for screenshots of errors or evasions. Since there are over 500K clusters, manual inspection seems extremely time consuming. How exactly was this task done?

- It is interesting that phishing websites tend to employ older versions of JavaScript libraries compared to newer versions used by the legitimate brand websites. Even though the primary goal of phishing websites is to lure victims by mimicing the original websites as closely as possible, why would phishing websites be reluctant to update their websites? Adding some discussions on this might be helpful.

- The added takeaway sections in each of the subsections are a nice touch.

**Reviewer Confidence:**

3: The reviewer is confident but not certain that the evaluation is correct

**Scope:**

4: The work is relevant to the Web and to the track, and is of broad interest to the community

---

### Official Review · Reviewer_fEKC · 2023-11-22

**Novelty:** 5
**Technical Quality:** 6

**Review:**

This paper analyzes the client-side resources (e.g., JavaScript libraries) used in phishing websites in comparison to the real websites of the target brands. The analysis is based on a large collection of phishing sites (7.1M phishing URLs). The analysis leads to recommendations for future defense.

Strengths:

-	The authors collected and analyzed large-scale real-world datasets.
-	New insights into phishing sites’ javascript usage
-	The dataset will be shared with the research community

Weaknesses

-	Overclaim on the size of the dataset
-	It would be nice to actually analyze phishing pages that handle 2FAs.
-	Did not evaluate any of the defense ideas.


This is a very interesting paper with extensive data collection efforts over 2 years. The analysis shows various distinct characteristics of client-side Javascript library usage in phishing sites. The most interesting result (to me) is the libraries that are exclusively used by phishing sites.

Section 4: What kind of IP addresses are used for data crawling (university IPs, cloud services, etc)? It is well known that phishing websites will perform cloaking. Based on Table 1, more than half of the phishing URLs are not accessible, and another half of the remaining URL crawling contains an error code. I am not sure how many of them are due to active cloaking and how many are due to the fact that the phishing pages have been taken down before you can crawl them.

The number of phishing sites that actually contribute to the analysis is 3M (instead of 7M) because many phishing URLs are not accessible or contain errors. 3M is still a large number, but I think the introduction and abstract can do a better job by stating this more precisely. Otherwise, people might think the analysis results and conclusions are based on the analysis of 7M sites (misleading).

The 2FA comment seems a bit strange. Yes, there are 22% phishing pages that do not use Javascript. It is not too surprising that such simple phishing pages exist. Since 2FA is mentioned, in your collection, how often do you encounter phishing pages that handle 2FA and perform MITM? This could be a much more interesting analysis.

Can you further clarify the recommendations? “If suspicious URLs are included in referrers, CDN service providers may consider interrupting the provision of resources” This seems to be a risky thing to do for CDN as it can easily disrupt legitimate customers. Is it the CDN’s job to detect “suspicious URLs”? Can they do it accurately?

**Questions:**

Please see the questions at the end of the review above.

**Reviewer Confidence:**

4: The reviewer is certain that the evaluation is correct and very familiar with the relevant literature

**Scope:**

4: The work is relevant to the Web and to the track, and is of broad interest to the community

---

### Official Review · Reviewer_Lf59 · 2023-11-24

**Novelty:** 4
**Technical Quality:** 4

**Review:**

**Summary**


The paper focuses on the inclusion of client-side resources such as javascript packages in the phishing websites targeting top brand websites. The paper shows that the number of third-party javascript files in the phishing pages are significantly larger that the corresponding benign websites. The paper also discusses the versions and the places that those libraries are loaded from and shows phishing websites generally rely more on CDNs.



**strength**
 -  The paper is well-written. The contribution is clear and the structure is easy to follow
 - Research on the development of phishing websites and the software development aspects of those websites are interesting and insights of these sort can be used in the defense side.

** weaknesses**
 - It is not clear how the findings of the paper can be translated into defense insights.


**Detailed Comments**
The paper focuses on an interesting area in analyzing phishing websites. The pipeline includes a crawler to capture the dynamic aspect of the phishing websites. The paper discusses the differences in the resource inclusion of phishing websites when compared with the corresponding legitimate websites. The paper is well-written and offers some interesting insights of the development of phishing websites in the wild. That said, the paper can be improved in a few ways making the contribution more concise.

It would be interesting to check if the third-party javascripts in the phishing websites are simply inserted via script tag or they are loaded and executed. Since the authors are using the chrome debugging protocol they should be able to see what code was loaded and executed. In fact, the question is if those javascript libraries are serving a purpose or they are just part of the DOM with no evident impact on the page.

It would be also interesting to have a deeper analysis of the exact overlaps between phishing and legitimate websites. Often time what used to happen was that the phishing page was a copy of the target page with some added scripts to establish a backdoor with adversaries. It is not clear to me if this holds any more. How much overlap the authors observed between those cases? This was partially discussed but was there any quantitative analysis?


Also, it would be great if the authors discuss the key takeaways of the findings and how they can shape our understanding on how to defend against new forms of phishing attacks. The fact that a large number of phishing websites are loading their scripts from CDNs is interesting, but how this finding or other findings of the paper can be used in the defense side. It would be great if the authors formulate a more comprehensive discussion by putting the findings into the context of phishing defense.

**Questions:**

Please review the comments.

**Ethics Review Description:**

No ethics concern was observed.

**Reviewer Confidence:**

4: The reviewer is certain that the evaluation is correct and very familiar with the relevant literature

**Scope:**

4: The work is relevant to the Web and to the track, and is of broad interest to the community

---

### Official Review · Reviewer_D4eh · 2023-11-27

**Novelty:** 4
**Technical Quality:** 5

**Review:**

This paper provides a study on how client-side resources (such as JavaScript libraries) are used in phishing websites in comparison to the legitimate target websites. The authors analyzed 7.1M phishing websites over 25 months. The findings include the fact that phishing websites tend to employ more diverse JavaScript libraries than the legitimate sites do, however they also tend to use significantly older versions of the correspondent libraries. They also found examples of libraries used only by phishing websites (e.g., Clipboard.js or Socket.IO), and observed phishing websites with less JavaScript.

Pros:
- The paper provides a longitudinal study of the JavaScript resources included in phishing pages and provides a clear comparison with target websites.
- Some findings are interesting and can be used to improve detections of phishing pages in future research.
- The authors provide some interesting methods (such as phishing page clustering and HTML similarity matching)

Cons:
- Classic phishing attacks are implemented by copying target phishing pages using tools such as Gophish or SET, or by using phishing kits that provide target brand templates. For this study, I would expect analysis of the source code that phishers use, or at least some in-depth investigation of findings (e.g., whether inclusion of outdated JavaScript libraries is caused by specific phishing kits).
- As an outcome of this measurement study I would expect a discussion on how to use the findings to improve phishing detection, even if not a proof-of-the-concept evaluations.
- One of the main findings that phishing pages tend to use older JavaScript library versions requires more temporal analysis, i.e. whether the library was already old when the correspondents phishing campaign started. It is likely the case that in the beginning phishers take up-to-date snapshots of the target websites. Alternatively, you can elaborate in more detail how fresh are the analyzed phishing websites.
- Identifying target webpages and mimicked brands using HTML and style similarity is not a novel direction and needs more evaluation and comparison to prior works in order to be claimed as a contribution.
- Analysis of the other client-side resources is very limited (section 8 is short and does not mention other images, json and data files, etc.)
- Technical analysis does not distinguish between bundled libraries together, or cases when same libraries are copied as inline JavaScript.

Smaller notes:
- It is unclear why the authors decided to use different crawlers to collect phishing pages and benign websites. This may introduce some differences as websites may cloak differently against Internet Archive bots and the customer crawler.
- The authors mention sophisticated phishing pages that can bypass 2FA in several places (such as section 6.2), but the context does not look correct as I do not think that 2FA bypass implies extensive usage of JavaScript or complex web pages structure.
- I was surprised not to find AES decryption libraries in table 2 as there are large phishing campaigns that unpack content with “Aes.Ctr.decrypt” and write to the page.

In general, I feel like more work could be done to have a more complete story and position the paper more strongly.

Update After Rebuttal
Thank you for the detailed answers. The additional results definitely strengthen the paper's contributions.

**Questions:**

- Can the take-aways identified by this paper be used to improve phishing page detection?
- Have you looked at the source codes that phishers use, such as collections of phishing kits and whether those match with you findings?
- How fresh the phishing websites that you crawled? Have you seen the same campaign reappear that can explain usage of older libraries?
- Could you describe more on how you analyze inline JavaScript and how do you process bundled libraries in a single file?

**Reviewer Confidence:**

4: The reviewer is certain that the evaluation is correct and very familiar with the relevant literature

**Scope:**

4: The work is relevant to the Web and to the track, and is of broad interest to the community

---

### Decision · Program_Chairs · 2024-01-22

**Decision:**

Accept (Oral)

**Comment:**

# Summary

 This paper presents a longitudinal study of phishing websites over 25 months. A key aspect that is studied is the reliance/inclusion of JavaScript in phishing websites vs. brand websites. The results show that this is a potential intriguing avenue for detection of phishing websites.

 # Strengths

 + Novel longitudinal study of the JavaScript resources included in phishing pages and provides a clear comparison with target websites.
 + The paper is well-written.
 + Insights from the analysis can be used to detect phishing websites.

 # Weaknesses

 - No temporal analysis of inclusions. (Note: authors described this experiment and the results in the rebuttal, and it will be included in the final version).
 - Overclaim on the size of the dataset (Note: authors will fix).

 # Recommendation

 Based on the discussion with the authors, it is clear that the authors can address most of the reviewers' concerns regarding the weaknesses of this paper. Given the novelty of the research and the value insights into the nature of phishing, this paper should be accepted to TheWebConf.

 ---